# Current Concepts in the Resection of Bone Tumors Using a Patient-Specific Three-Dimensional Printed Cutting Guide

Hisaki Aiba [1,2], Benedetta Spazzoli [1], Shinji Tsukamoto [3], Andreas F. Mavrogenis [4], Tomas Hermann [1,5], Hiroaki Kimura [2], Hideki Murakami [2], Davide Maria Donati [1] and Costantino Errani [1,*]

1    Department of Orthopedic Oncology, IRCCS Istituto Ortopedico Rizzoli, Via Pupilli 1, 40136 Bologna, Italy
2    Department of Orthopedic Surgery, Nagoya City University, Nagoya 467-8601, Aichi, Japan
3    Department of Orthopedic Surgery, Nara Medical University, Kashihara 634-8521, Nara, Japan
4    First Department of Orthopedics, School of Medicine, National and Kapodistrian University of Athens, 11527 Athens, Greece
5    Department of Tumors, HTC Hospital, Traumagologico Concepcion, 1580 San Martin, Concepcion 4030000, Chile
*    Correspondence: costantino.errani@ior.it; Tel.: +39-051-6366103

**Abstract:** Orthopedic oncology has begun to use three-dimensional-printing technology, which is expected to improve the accuracy of osteotomies, ensure a safe margin, and facilitate precise surgery. However, several difficulties should be considered. Cadaver and clinical studies have reported more accurate osteotomies for bone-tumor resection using patient-specific cutting guides, especially in challenging areas such as the sacrum and pelvis, compared to manual osteotomies. Patient-specific cutting guides can help surgeons achieve resection with negative margins and reduce blood loss and operating time. Furthermore, this patient-specific cutting guide could be combined with more precise reconstruction using patient-specific implants or massive bone allografts. This review provides an overview of the basic technologies used in the production of patient-specific cutting guides and discusses their current status, advantages, and limitations. Moreover, we summarize cadaveric and clinical studies on the use of these guides in orthopedic oncology.

**Keywords:** three-dimensional printed guide; patients-specific guide; bone tumor; limb-sparing surgery; orthopedic oncology; pelvic tumor; tumor of the sacrum; patient-specific implant

## 1. Introduction

Three-dimensional (3D)-printing technology has been introduced in the field of orthopedic surgery, including its application in total joint arthroplasty, the treatment of malunion, and the reconstruction of bone defects due to trauma or bone tumors [1–3]. Treating complex comminuted fractures requires extensive surgical experience and anatomical knowledge, and patients with malunion could require osteotomy to correct the deformity and relieve pain [4]. 3D-printing technology could help precise and effective fracture fixation and reconstruction [4]. In orthopedic oncology, with the aim of limb-sparing surgery, saving normal bone stock with adequate margins is an important process [5,6]. Due to the anatomical complexity of challenging sites, such as the pelvis or sacrum, the application of 3D-printing technology for precise resections of bone tumors may be an effective strategy [7,8]. The treatment of bone tumors within such locations is challenging because the precise recognition of the geometries of the pelvis or sacrum and the adjacent structures is difficult [9]. Even for experienced surgeons, consistent maintenance of the surgical margin during pelvic surgery remains challenging [5]. Patient-specific cutting guides (PSGs) have recently been used to facilitate surgeons in more precise planning of surgical interventions [6,10,11]. In this article, we review the basic technology for the production of PSGs and its current status, highlighting its advantages and limitations in the field of orthopedic oncology.

## 2. Digital Imaging Applications for Surgical Interventions

### 2.1. Reproduction of 3D Images

Using Digital Imaging and Communication in Medicine approaches (DICOM), the 3D images of tumors with normal anatomy are reconstructed using segmented thin-slice computed tomography (CT) and magnetic resonance imaging (MRI) [11]. The reconstructed anatomical 3D images of bony tissues and tumors are then exported as a series of polygons, the number of which directly correlates with the resolution in the standard triangulation language format [12]. The files then undergo processing using several available software packages to build lighter meshes to reduce the unnecessary computational load or shorten the production time during the computer-aided design phase [12].

### 2.2. Computer-Aided Surgical Simulation

Reconstructed 3D images are often exported to other software for the design of PSGs. Creo Parametrics (Parametric Technology Corp.; Needham, MA, USA), 3D Studio Max (Autodesk, San Rafael, CA, USA), and MIMICS (Materialize, Leuven, Belgium) are widely used software providers [13]. After confirming the geometric measurements of the targeted bone, the internal surface of the PSG is designed to perfectly fit the cortical bone of the patient. Surgeons and engineers can determine the cutting position via discussions and then define several slits and holes for sawblade insertion and fixation pins while cutting [14,15]. In addition, a contact-surface widening technique or deformable clip can be used [13]. Subsequently, several printing techniques are used, including stereolithography, fused deposition modeling, and selective laser sintering [7]. Due to the accuracy and versatility of available materials, additive manufacturing techniques are typically preferred for medical applications over subtractive manufacturing [16,17].

## 3. 3D-printing Technique for Surgical Planning

### 3.1. Advantages of the 3D-Printing Technique

#### 3.1.1. Resection with Safe Margins

One of the most important advantages of PSGs is the achievement of an accurate resection of bone tumors with safe margins [11,18,19]. From previous cadaver studies, manual resection can result in inaccuracies of up to 5–15 mm, which might lead to unexpected intralesional resection [20–23]. A PSG can be applied for the preparation of massive bone allografts (MBAs) for the reconstruction of bone defects following bone-tumor resection (Figure 1) [24]. A PSG can help surgeons achieve a more precise cutting of the MBA using the same predefined section planes for the resection of bone tumors [24]. Bellanove et al. reported a case series of four patients and the outcomes of the resection of a malignant bone tumor in the proximal tibia using a PSG and MBA [25]. Safe resection margins were achieved, and satisfactory postoperative radiographs were obtained [25]. In addition, radiological union at the graft—host junction was observed at 4–12 months [25].

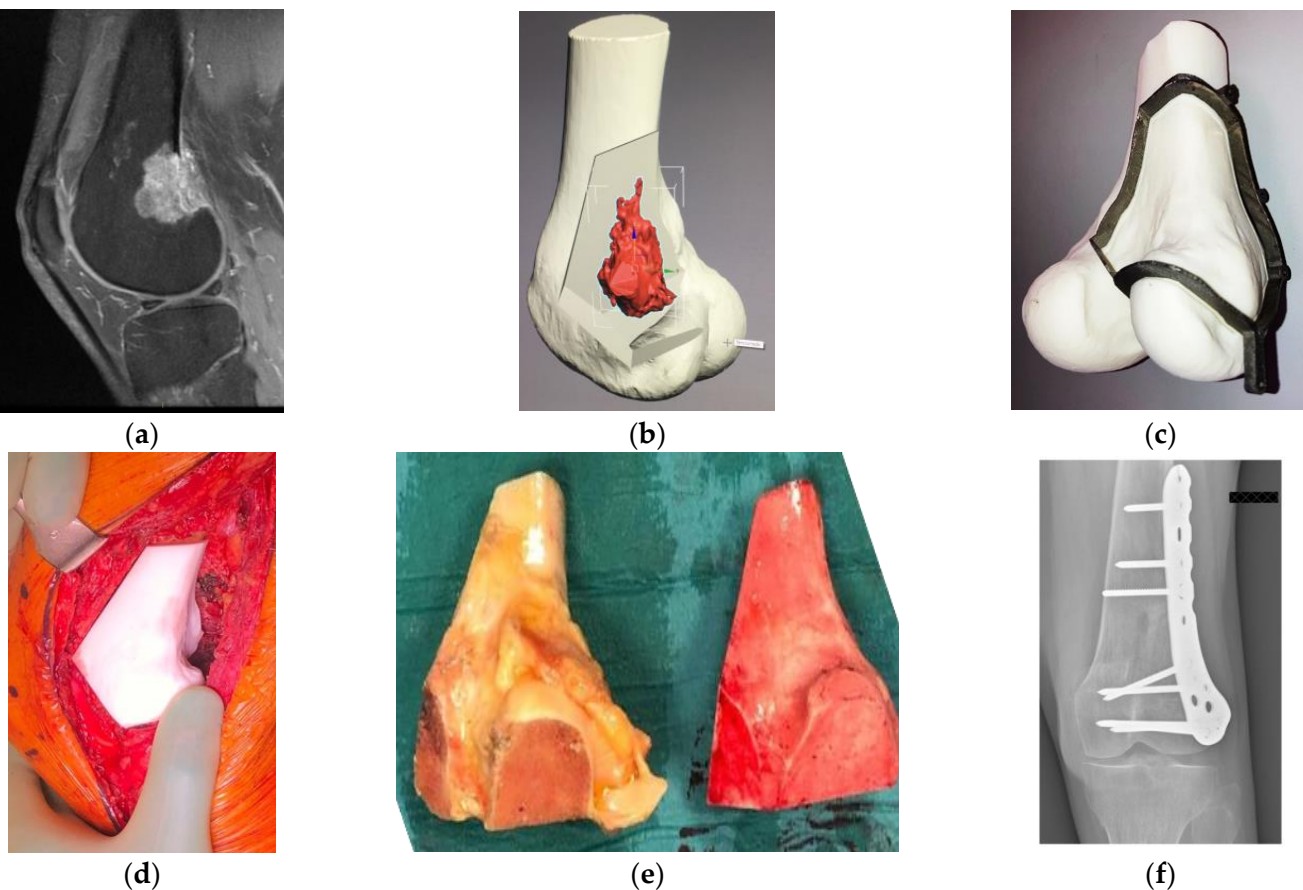

**Figure 1.** Osteosarcoma in the distal femur of a 56-year-old man. (**a**) High-intensity signal was seen in the sagittal MRI image (T1-fat suppression with contrast) at the posterior to the lateral side of the distal femur. (**b**) Resection planning in the computer-aided design phase. The location of the tumor was highlighted in red via an integrated image of CT and MRI and resection planes were determined with a discussion between surgeons and engineers. (**c**) Surgical planning with a PSG and an artificial bone. To preserve the surface of the knee joint and bone stock, hemi-cortical resection with the PSG, followed by reconstruction with a massive bone allograft was planned. The fitting of the PSG to the bone was confirmed. (**d**) Intraoperatively, after the resection with the PSG, the compatibility of the bone defect with the allograft, which was subsequently resected using the PSG, was confirmed by the artificial bone spacer. (**e**) Resected specimen (left) and resected massive bone allograft (right) via the PSG. We confirmed the identical fitting of a massive bone allograft to the large defect after the tumor resection. (**f**) Postoperative radiograph after the resection of the tumor and reconstruction with massive bone allograft. CT, computed tomography; PSG, patient-specific cutting guide; MRI, magnetic resonance imaging.

### 3.1.2. Reconstruction of Bone Defects

The design of conventional prosthetic implants has improved, although, it is important to consider the potential failure of initial fixation due to inadequate matching of implants and host bone tissue, which affects the bone–implant interaction, leading to bone atrophy or implant loosening [16,26]. A patient-specific implant (PSI) is used to ensure a good fit for bone defects for accurate placement of prostheses; however, its clinical utility should be validated over the long term [19] (Figure 2). Kieser et al. reported mid-term outcomes (a median follow-up period of 38 months) of a PSI for large bone defects in the acetabular region [27]. Of the 36 patients evaluated, one patient experienced early implant migration with subsequent stabilization; two patients experienced failure of osteointegration; and no patient exhibited aseptic loosening [27]. Liu et al. reported the results of a retrospective study of a P2–P3 resection of pelvic tumors, with a median follow-up period of

36 months [28]. Among the 19 patients treated with PSG + PSI, aseptic loosening occurred in four patients [28] (Figure 2).

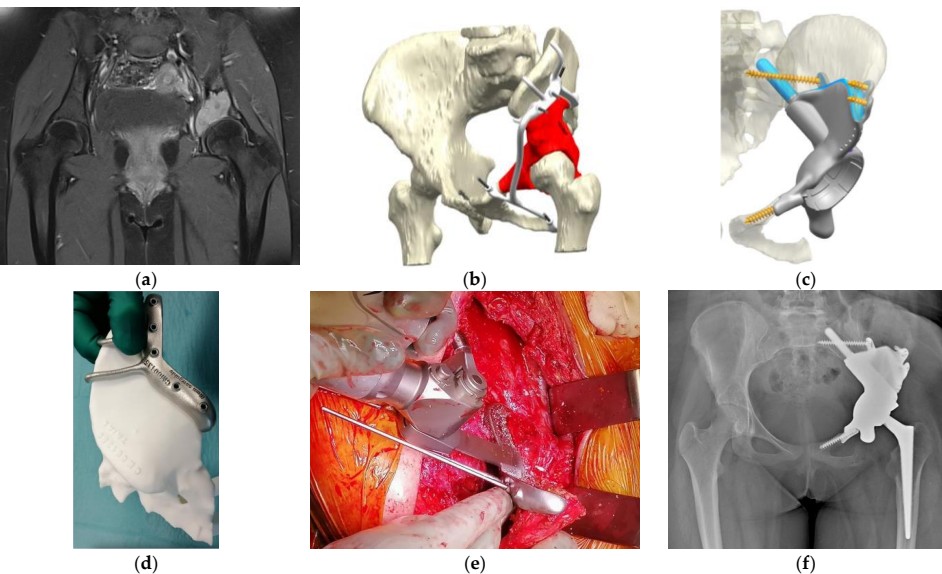

(a)   (b)   (c)

(d)   (e)   (f)

**Figure 2.** Osteosarcoma in the pelvis in a 14-year-old girl. (**a**) MRI image (T1-fat suppression with contrast) shows an osteolytic lesion in the left acetabulum with a high-intensity signal. (**b**) Preoperative planning for the P2–P3 resection of the pelvis using a PSG and reconstruction using a PSI. The tumor's location was highlighted in red. The PSG was designed to have a sufficient margin with hooks for better attachment of the pelvic bone. The PSG was segmented for detachment into small parts, enabling a step-by-step surgical procedure. (**c**) In the computer-aided-design phase, the insertion of the PSI after the resection of the tumor and the points of fixation of the PSI were simulated. The actual cutting points were confirmed with a 3D-printed bone model. The points of fixation of cancellous screws and stems were also determined, and the appropriate length of drilling or the sizes of planned screws or stems were calculated. (**d**) The actual cutting points were preoperatively simulated with an artificial, 3D-printed bone model. (**e**) An intraoperative image of the resection along with the PSG. This procedure was supported by Waldemar Link GmbH & Co. KG (Norderstedt, Germany). (**f**) Postoperative radiograph after the resection of the tumor and reconstruction with a PSI. PSI, patient-specific implant; PSG, patient-specific cutting guide; MRI, magnetic resonance imaging.

### 3.1.3. Understanding of Anatomy and Surgical Planning

Surgical planning is important for successful outcomes at challenging sites such as the pelvis and sacrum. This planning is improved with a better understanding of the spatial relationship between important adjacent structures and tumors [29,30]. By using a sterile 3D-printed model during the operation, proper orientation of the anatomy can be provided for surgeons and assistants [7,31]. Goyal et al. described the utility of education in orthopedic residents using a 3D-printed model [32]. They compared education with lectures only (Group 1) or lectures with 3D-printed-model guidance (Group 2). The post-test scores for fracture classification and surgical approach were significantly higher in the 3D model group (Group 1 vs. 2: 2.5–6 vs. 4.4–10%; $p < 0.05$) [32]. With a better and more accurate understanding of anatomy, surgeons are confident in maintaining a safe surgical margin without compromising important structures [33]. The educational approach, when combined with the 3D model and the PSG, enables precise control of safe margins by demonstrating osteotomies with oscillating blades in the predefined resection planes to young surgeons [24].

### 3.1.4. Reduction of Surgical Invasiveness and Operation Time

Several studies have shown that PSGs can reduce blood loss and operation time during bone tumor surgery [8,34]. In a retrospective comparative study, Liu et al. conducted a retrospective comparative study of patients with malignant bone tumors in the pelvis (P2–P3) treated with or without PSGs (*n* = 19/19), and they found that operation time (PSG group vs. control group: 209 vs. 272 min; *p* = 0.003) and blood loss (PSG group vs. control group: 1390 vs. 2248 mL; *p* = 0.002) were better in the PSG group [28]. In contrast, a randomized control study by Wang et al. described the efficacy of the PSG for the resection of malignant bone tumors around the knee joint (*n* = 33, control group; *n* = 33, PSG group) [34]. They proved the superiority of the PSG in terms of blood loss (control group vs. PSG group: 689 vs. 647 mL; *p* = 0.003) [34], but no difference was observed in operation time (control group vs. PSG group: 136 vs. 145 min; *p* = 0.685) [34].

The PSG is expected to enable surgeons to reproduce a virtual surgical plan with similar accuracy but with less bone resection time when compared with navigation assistance during surgery [35]. Bosma et al. compared PSGs and navigation-assisted osteotomy for knee-joint resection in a cadaver study that involved 16 simulated tumors around the knee in four cadavers [22]. The PSG group had a significantly lower total bone resection time (navigation group vs. PSG group: 17 vs. 5 min; *p* < 0.001), and the maximum distance between the planned osteotomy and the achieved osteotomy was superior in the PSG group (PSG group vs. navigation-assisted group: 1.9 vs. 3.6 mm, *p* = 0.042) [22].

### 3.2. Limitations of the 3D-Printing Technique

#### 3.2.1. Delay and Cost of Surgery

Lead time is required for the design and manufacturing of PSGs. The intervals between diagnosis and surgery are related to the diagnosis of the bone tumor—longer intervals in chemo-sensitive tumors, such as osteosarcoma or Ewing sarcoma; shorter intervals in chemo-resistant tumors, such as chondrosarcoma [36]. If there was adequate time before surgery, it would be possible to prepare the PSG [36]. With the advances in technology, any delay during preparation has been shortened, and Rustemeyer et al. mentioned a 2–4-week delay in PSG- or PSI-assisted surgeries for maxillofacial applications [37]. However, Martelli et al. reported that production delays were a limitation in 19.6% of studies that used 3D-printed devices [38]. Outsourcing the production of a PSG or PSI could take 4–6 weeks for the process of design and manufacturing [39]; whereas, with the development of 3D printing devices, low-cost in-house 3D-printing technology has been utilized, and the introduction of PSGs is becoming more simplified [14]. Calvo-Haro et al. reported that the average working hours for processing were 12 h and the operating time for 3D printing was 10 h based on their experience with the manufacturing process of the anatomical model, or PSG, for a total of 623 orthopedic surgeries in a single institution [40]. They also reported that the proficiency or complexity of the model might influence the time taken for the process [40]. Frizziero et al. reported that PSGs designed for pediatric orthopedic femoral osteotomies can be provided for approximately EUR 300 in 1.5 h (printing time) with a low-end fused deposition modeling 3D printer [14]. Although the cost of 3D printing is influenced by the size of the model, the material used, and the quality of the resolution, Fidanza et al. found that the production cost for anatomical full-touch real-size bio-models is less than EUR 5 per model for material (polylactic acid) and less than EUR 1000 for hardware installation [41].

#### 3.2.2. Learning Curves

Unlike the navigation system, PSGs cannot provide accurate feedback on the patient's position during surgery [23,36,42]. This inaccurate process may result in an unexpected deviation of the PSG setting from the planned correct position without consciousness. Thus, the subjective feeling of fitting the bony contour is important. Methods to verify the intraoperative position of the PSG should be developed [43]. Wong et al. first reported the results of a combined technique with a PSG and a navigation technique in three pa-

tients with primary bone tumors around the knee joints [44]. The combination technique revealed that the mean maximum deviation errors in osteotomy were 1.64 ± 0.35 mm [44]. Considering the accuracy of the navigation system of 2.43–3.60 mm in previous studies [22,43,45,46], this combination technique in joint-preserving tumor surgery suggested acceptable results [43,44].

Most software programs that facilitate the virtual process of PSG installation do not consider the soft tissues surrounding the surgical site [13]. In the real world, an installation should be planned considering the ligaments, cartilage, tendons, and muscles [13]. Mustahsan et al. reported a cadaver comparison study, which simulated wide resections of a bone sarcoma on 24 cadaver femurs, with and without soft tissue coverage, and concluded that soft tissue coverage caused random positioning errors of PSGs without a spike shape (smooth vs. soft tissue covered: 5.0 vs. 6.5 mm, mean deviation of the cutting planes from the planned plane) [47]. Furthermore, Dong et al. reported a retrospective study of PSG-guided malignant bone-tumor resection (pelvic = 10, femoral = 4, and tibia = 3; 64 osteotomies of tumor-affected bone) [7]. Although 63 of 64 (98%) osteotomies achieved wide resection and negative margins, one osteotomy had a contaminated margin due to the unexpected swing of the saw and flexibility of the guide [7]. The basic skills of orthopedic surgeons remain important for appropriate exposure of the surrounding tissue and accurate osteotomies [48].

Currently, preoperative planning can be achieved in a more realistic manner using a haptic component [13,31]. This technology allows surgeons to simulate the actual procedure with the tactile sensation of nearby soft tissue and the placement of the PSG [13,31]. Additionally, several optical feedback systems, including augmented reality or intraoperative reference of preoperative images, are feasible tools for correcting PSG positioning and improving the accuracy of bone cuts [47,49]. These new technologies may aid orthopedic surgeons in their learning curve.

### 3.2.3. Properties of Surgical Materials

Metallic or non-metallic materials can be used in 3D-printing technology [4]. The safety of these materials should be ensured in accordance with national and industrial standards [4]. Metallic guides are sufficiently strong to prevent unexpected chips or guide deformities during surgery using a surgical saw or electronic heating tools [4,50]. However, the running cost for the preparation of metallic materials is higher because the modeling technology should utilize selective laser melting or electron-beam melting [51].

In the case of non-metallic materials, material characteristics should be considered before their use in the surgical field. A PSG must be sterilized without compromising its mechanical resistance and design geometry [14]. As the PSG is administered directly into the bone, the risk of infection should be closely monitored [52]. Autoclaving is generally used as a sterilization method [14]. Thus, tolerance to aggressive steam heat cycles should be noted. High-molecular polymers, such as acrylonitrile butadiene styrene, polyethylene terephthalate glycol, and simple polylactic acid, cannot withstand temperatures above 50 °C without significant loss of mechanical properties [53]. As an alternative sterilization method, ethylene oxide at 37 °C for 16 h may be used [53]. Conversely, polyether-ketone has thermomechanical resistance and biocompatibility but at a higher cost [54]. For non-metallic materials, high-temperature polylactic acid may be used because it is printable with fused deposition modeling and has the capacity for aggressive steam heat cycles while maintaining the same designed geometry [14,55].

### *3.3. Validation Studies*

#### 3.3.1. Cadaver Studies

Many cadaver studies have been conducted investigating PSG resection during actual surgical procedures (Table 1). Regarding resection of the femur, Khan et al. demonstrated the superiority of PSG resection for the hemi-metaphyseal resection of primary bone tumor models in the distal femur (six matched pairs of cadaveric femurs). They stated

that the maximal deviations from the preoperative plan were 9.0 mm vs. 2.0 mm in the manual group and the PSG group, respectively ($p = 0.002$), while the maximum deviation from planned osteotomy lines greater than 3 mm occurred in all patients in the manual group and 5.6% of patients in the PSG group [20]. Helguero et al. reported box-shaped hemi-metaphyseal osteotomies (three planes: top, middle, and bottom) in the distal femur resections using PSGs (nine matched pairs of cadaveric femurs) [21]. They compared the deviation from the planned angles in several planes: the top plane, 7.16° vs. 5.30° ($p = 0.358$); the middle plane, 4.41° vs. 1.78° ($p = 0.038$); and the bottom plane, 7.96° vs. 2.20° ($p = 0.003$), in the manual resection group and the PSG group, respectively [21]. Bosma et al. compared the maximum distance between the planned osteotomy and the achieved osteotomy of the distal femur and proximal tibia by comparing manual resection, PSG resection, navigation techniques, or PSG resection + navigation techniques using 16 simulated tumors around the knee in four human cadavers [22]. The distance between the planned osteotomy and the achieved osteotomy was 9.2 mm in the manual group, 3.6 mm in the navigation group, 1.9 mm in the PSG group, and 2.0 mm in the PSG + navigation group [22]. The distance between the planned osteotomy and the achieved osteotomy between the PSG + navigation group and the PSG group had no significant differences ($p = 0.92$). However, compared to the navigation group, both PSG + navigation ($p = 0.042$) and PSG resections ($p = 0.034$) showed significantly higher accuracy in the distance between the planned osteotomy and the achieved osteotomy [22]. Sallent et al. compared the accuracy of osteotomies at different points of the pelvis via PSG resection or manual resection in five cadaveric pelvic bones [23]. They analyzed the deviation of osteotomies from the preoperative plan and demonstrated that the PSG resection was superior to manual resection at all points (sacroiliac area (14.6 mm in manual resection vs. 5.0 mm in PSG resection, $p = 0.008$), supra-acetabular area (10.2 mm in manual resection vs. 4.0 mm in PSG resection, $p = 0.008$), ischial area (5.20 mm in manual resection vs. 2.2 mm in PSG resection, $p = 0.016$), and iliopubic area (3.00 mm in manual resection vs. 0.8 mm in PSG resection, $p = 0.032$)) [23].

**Table 1.** Cadaver study reports of bone resection/manual resection versus navigation resection vs. PSG resection.

| Authors | Site | Technology | Evaluation | Results |
|---|---|---|---|---|
| Khan et al. [20] | Femur | MAN vs. PSG | Deviation from the planned osteotomy line | 9.0 mm (MAN) vs. 2.0 mm (PSG), $p = 0.002$ |
| Helguero et al. [21] | Femur | MAN vs. PSG | Deviation from the planned osteotomy angle | MAN: large gaps (>5 mm) between the implant and the bone PSG: no large gaps, no statistics |
| Bosma et al. [22] | Femur, tibia | MAN vs. NVI vs. PSG vs. NVI + PSG | Deviation from the planned osteotomy line | 9.2 mm (MAN), 3.6 mm (NVI), 1.9 mm (PSG), and 2.0 mm (NVI + PSG) [*1] |
| Sallent et al. [23] | Pelvis | MAN vs. PSG | Deviation from the planned osteotomy line | Sacroiliac: 14.6 mm (MAN) vs. 5.0 mm (PSG), $p = 0.008$ Supra-acetabular: 10.2 mm (MAN) vs. 4.0 mm (PSG), $p = 0.008$ Ischial: 5.2 mm (MAN) vs. 2.2 mm (PSG), $p = 0.016$ Iliopubic: 3.0 mm (MAN) vs. 0.8 mm (PSG), $p = 0.032$ |
| García-Sevilla et al. [18] | Pelvis | PSG | Translations and rotations of the planned osteotomy plane | Iliac crest: mean translation, 5.3 mm; mean rotation, 6.7° Supra-acetabular: mean translation, 1.8 mm; mean rotation, 5.1° Ischial: mean translation, 1.5 mm; mean rotation, 3.4° Pubic: mean translation, 1.8 mm; mean rotation, 3.5° |

MAN, manual cutting; NVI, navigation guide cutting; PSG, patient-specific guide; SD, standard deviation. [*1]. The MAN group was significantly less accurate than the other groups ($p < 0.001$). No significant difference was observed in location accuracy between the PSG and NVI + PSG groups ($p = 0.92$). The PSG and NVI + PSG groups were significantly more accurate than the NVI group ($p = 0.034$ and $p = 0.042$, respectively).

### 3.3.2. Clinical Application

PSGs may be indicated for anatomically challenging sites [2,10,34], the requirement of precise osteotomy in combination with a PSI or MBA [6,8,34,56,57], and the preservation of joint or bone stock with precise or complex osteotomy [20,21].

In 2014, Gouin et al. published a case series of 11 patients with pelvic bone tumors who underwent resection using PSGs [10]. They reported that all patients had negative margins, with a minimum distance of 2.5 mm between the achieved osteotomies and the tumor boundary; the mean cutting error was 0.8 mm [10]. Furthermore, Evrard et al. analyzed the correlation between planned and obtained margins (excellent correlation, $R^2$ = 0.841; $p$ < 0.0001) from the bone-tumor resections in various locations (n = 31) [24].

Several studies explored the combination of PSG resection with MBA or PSI reconstructions [6,8,34,56,57]. Ma et al. described eight patients who underwent resection of the bone tumor via a PSG in the distal femur and reconstruction with an MBA. They reported precise resection with negative margins for all patients and good outcomes with MSTS scores ranging from 70–100% and a mean knee flexion of 112.5° [8]. Similarly, Müller et al. reported the resection of tumors in various bones (scapula = 4, pelvis = 3, tibia = 3, femur = 2) using a PSG and reconstruction with an MBA. They observed a range of deviations of osteotomies from planned cutting lines of 0.74–3.60 mm [7]. In another study, Liu et al. reported on 12 patients with metaphyseal malignant bone tumors around the knee who underwent resection using a PSG and intercalary PSI reconstruction [58]. They found that all patients achieved negative margins with the accurate matching of residual bone and intercalary PSI; the mean MSTS score was 28 points [58]. Dong et al. retrospectively reported the treatment of bone tumors in the extremities ($n$ = 7) and pelvis ($n$ = 10) using a PSG with an MBA and/or autograft [7]. They stated that 98% of osteotomies achieved a negative margin, and biological reconstruction showed a good bone healing rate of 91.7% [6]. Finally, Hu et al. reported a combination technique using PSGs in the resection of bone tumors in the shoulder and reconstruction with reverse shoulder arthroplasty and a 3D-printed glenoid prosthesis. They analyzed seven patients with a bone tumor in the proximal humerus and found that negative margins were achieved in all patients with good postoperative function (mean MSTS score = 85.7%) and no instability or scapular notching (average osteotomy length = 118.6 mm) [30].

Several studies have compared the results of bone-tumor resection using PSGs [28,34,56]. In a prospective randomized control study ($n$ = 33, conventional tumor resection group (control); $n$ = 33, PSG group), Wang et al. reported the superiority of PSGs in osteotomies of bone tumors around knee joints in terms of blood loss (689 mL (control) vs. 650 mL (PSG), $p$ = 0.037) and postoperative function (mean MSTS score = 26.2 (control) vs. 28.3 (PSG), $p$ = 0.019). However, there was no significant difference between margin status (90.9% (control) vs. 93.9% (PSG)) or operation time (136 min (control) vs. 145 min (PSG), $p$ = 0.685) [34]. In a retrospective study comparing 19 patients with pelvic tumors treated with manual resection and nine patients with pelvic bone tumors treated via PSGs, Evrard et al. reported the superiority of the PSG group in local recurrence rate (37% (control) vs. 0% (PSG), $p$ = 0.035), margin status (68% (control) vs. 89% (PSG) group, $p$ = 0.479), and operation time (633 min (control) vs. 612 min (PSG), $p$ = 0.877) [56]. In a retrospective case-control study of P2–P3 resection of pelvic tumors including 19 patients treated with manual resection (control group) and 19 patients treated using PSG, Liu et al. reported the superiority of the PSG group; lower local recurrence rate (42% (control) vs. 5% (PSG), $p$ = 0.008); shorter operation time (272 min (control) vs. 209 min (PSG), $p$ = 0.002); less blood loss (2,248 mL (control) vs. 1,390 mL (PSG), $p$ = 0.002); and negative surgical margin (89.4% (control) vs. 100% (PSG), no statistical analysis) [28] (Table 2).

**Table 2.** Clinical studies investigating PSGs for the resection of bone tumors.

| Authors and Study Type | Tumor | Site | Patient Number | Surgical Technique | Negative Surgical Margin | Blood Loss (Mean) | Operation Time (Mean) | Local Recurrence | Accuracy of Osteotomy |
|---|---|---|---|---|---|---|---|---|---|
| Gouin et al. [10], case series | CS, EWS, SS | Pelvis | 11 | PSG | 100% | NA | NA | 9% | Mean cutting error * = 0.8 mm |
| Ma et al. [8], case series | OS | Femur | 8 | PSG + ALO | NA | 746 mL | 213 min | 0% | NA |
| Wang et al. [34], randomized control study | CS, GCT, OS | Femur, tibia | 33 | PSG + ALO | 90.9% (CTR) vs. 93.9% (PSG); NS | 689 mL (CTR) vs. 650 mL (PSG); $p = 0.037$ | 136 min (CTR) vs. 145 min (PSG); $p = 0.685$ | 15.2% (CTR) vs. 9.1% (PSG); $p = 0.708$ | NA |
| Evrard et al. [56], case-control study | CS, EWS, OS | Pelvis | 9 | PSG + ALO | 68.4% (CTR) vs. 89% (PSG); $p = 0.479$ | NA | 633 min (CTR) vs. 612 min (PSG); $p = 0.877$ | 37% (CTR) vs. 0% (PSG); $p = 0.035$ | NA |
| Park et al. [57], case series | CS, Meta, OS, SS | Various | 12 | PSG + PSI or ALO | 100% | NA | 118 min | 8.3% | Maximal cutting error = 3 mm |
| Hu et al. [30], case series | CS, GCT, OS | Shoulder | 7 | PSG + PSP + RSA | 100% | NA | NA | 0% | NA |
| Liu et al. [28], case-control study | CS, EWS, SS | Pelvis | 19 | PSG + PSI | 89.4% (CTR) vs. 100% (PSG), $p = $ NA | 2,248 mL (CTR) vs. 1,390 mL (PSG); $p = 0.002$ | 272 min (CTR) vs. 209 min (PSG); $p = 0.002$ | 42% (CTR) vs. 5% (PSG); $p = 0.008$ | 5 mm deviation from the planned margin, 58% (CTR) vs. 0% (PSG), $p = $ NA |
| Müller et al. [6], case series | CS, EWS, OS | Various | 12 | PSG + ALO | 92% | NA | NA | 0% | Range of cutting error = 0.7–3.6 mm |
| Liu et al. [58], case series | CS, EWS, SS | Femur, tibia (intercalary) | 19 | PSG + PSI | 100% | NA | 155 min | 8.3% | Mean cutting error = 1.9 mm |
| Wong et al. [44], case series | EWS, OS | Femur, tibia | 3 | PSG + NVI | 100% | NA | 276 min | 0% | Mean cutting error = 1.6 mm |
| Evrard et al. [24], case series | ADA, CS, EWS, FD GCT, OS, SS | Various | 31 | PSG | 100% | NA | NA | NA | Mean cutting error = 0.4 mm |
| Dong et al. [7], case series | EWS, Meta, OS, CS, GCT | Pelvis, femur, tibia | 17 | PSG + ALO or AUTO | 98% | Pelvis, 2,100 mL Limb, 957 mL | 618 min | 0% | NA |

* Mean cutting error indicates the distance between the planned and actual resection lines. ADA, adamantinoma; ALO, allograft; AUTO, autograft; CTR, control group; CS, chondrosarcoma; EWS, Ewing sarcoma; FD, fibrous dysplasia; GCT, giant cell tumor; MAN, manual cutting; Meta, metastatic tumors; min; minute; OS, osteosarcoma; NA, not analyzed; NVI, navigation guide cutting; NS, not significant; PSG, patient-specific guide; PSI, patient-specific implant; RSA, reverse shoulder arthroplasty; SS, soft-tissue sarcomas.

## 4. Conclusions

In this review, we report recent findings on the clinical utility of PSGs for bone-tumor resection, along with their advantages and limitations. 3D-printing technology is still in the development phase; however, there is a clear upward trend in its application, as evidenced by the increasing number of published articles, and the application of this technology will continue to progress. The use of a customized surgical osteotomy guide may allow safe margins, reduce surgical time, and also allow accurate matching between the residual bone and the PSI or MBA. Multicenter prospective studies are needed to confirm these preliminary results on a larger scale.

**Author Contributions:** Conceptualization, H.A., D.M.D. and C.E.; methodology, H.A. and D.M.D.; software, H.A.; B.S and C.E.; validation, H.A., B.S., T.H., S.T., H.M. and C.E.; formal analysis, H.A. and C.E.; investigation, H.A.; resources, D.M.D. and C.E.; data curation, C.E.; writing—original draft preparation, H.A.; writing—review and editing, H.A., T.H., D.M.D., S.T., H.K., A.F.M., H.M. and C.E.; visualization, H.A.; supervision, D.M.D. and C.E.; project administration, D.M.D. and C.E.; funding acquisition, D.M.D. and C.E. All authors have read and agreed to the published version of the manuscript.

**Funding:** This study received no external funding.

**Informed Consent Statement:** Informed consent was obtained from all the subjects involved in the study. Written informed consent to publish this paper was obtained from the patients and his/her guardians.

**Acknowledgments:** The authors thank the patients and their families.

**Conflicts of Interest:** The authors declare no conflict of interest.

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
