# Peer review of "Current Concepts in the Resection of Bone Tumors Using a Patient-Specific Three-Dimensional Printed Cutting Guide"

_curroncol, doi:10.3390/curroncol30040292_

Round 1

Reviewer 1 Report

This review is comprehensively written and covers the entire topic in depth. 

One minor point to address: following the lines 170ff, the delay for manufacturing the guides is well underscored. The example provided relates -as indicated- to maxilla-facial surgery, but not to orthopaedic oncology. Because the volume for the latter is much smaller, it is usually not a focus of the companies and therefore it takes much longer for orthopaedic oncology cases. Please also separate the information for PSI and PSG.

One major point: please include a paragraph on indications. The reader should not get the impression, PSG has to be used now for all conditions. 

Author Response

Thank you for the thoughtful comments on our article. Based on the suggestions, we attempted to improve the quality of this review. Additionally, we obtained proofreading service for the grammatical check.

“This review is comprehensively written and covers the entire topic in depth. 

One minor point to address: following lines 170ff, the delay for manufacturing the guides is well underscored. The example provided relates -as indicated- to maxilla-facial surgery, but not to orthopedic oncology. Because the volume for the latter is much smaller, it is usually not a focus of the companies and therefore it takes much longer for orthopedic oncology cases. Please also separate the information for PSI and PSG.”

Thank you for your comment. As per your suggestions, the manufacturing process might be versatile based on the size, materials, or purposes. Especially if we outsource the process to companies, the delivery time might cause a delay. We carefully describe this point as follows:

[original (line 175–180)]

However, Martelli et al. reported that 19.6% of studies using 3D-printed devices found production delays to be a limitation [38]. With the development of 3D printing devices, low-cost in-house 3D printing technology has been utilized, and the introduction of PSG is becoming more simplified [14]. By disregarding the time required for the computer-aided design phase, the initial costs for installation of related devices, or the limitations of the material properties, PSG can be provided for approximately 300 euros in 1.5 hours (printing time) with a low-end fused deposition modeling 3D printer [14].

[new]

With the advances in technology, any delay during preparation has been shortened, and Rustemeyer et al. mentioned a 2–4-week delay in PSG- or PSI-assisted surgeries for maxillofacial applications [38]. However, Martelli et al. reported that production delays were a limitation in 19.6% of studies that used 3D-printed devices [39]. Outsourcing the production of PSG or PSI could take 4–6 weeks for the process of design and manufacturing [40]; whereas, with the development of 3D printing devices, low-cost in-house 3D printing technology has been utilized, and the introduction of PSG is becoming more simplified [15]. Calvo-Haro et al. reported that the average working hours for processing were 12 hours and the operating time for 3D printing was 10 hours based on their experience with the manufacturing process of the anatomical model, or PSG, for a total of 623 orthopedic surgeries in a single institution [41]. They also reported that the proficiency or complexity of the model might influence the time taken for the process [41]. Frizziero et al. reported that PSG designed for pediatric orthopedic femoral osteotomies can be provided for approximately 300 euros in 1.5 hours (printing time) with a low-end fused deposition modeling 3D printer [15]. Although the cost of 3D printing is influenced by the sizes of the models, the material used, and the quality of the resolution, Fidanza et al. found that the production cost for anatomical full-touch real-size bio-models is less than 5 euros per model for material (polylactic acid) and less than 1,000 euros for hardware installation [42].

“One major point: please include a paragraph on indications. The reader should not get the impression PSG has to be used now for all conditions. “

Thanks for your opinion. We totally agree with you and understand that PSG should not be indicated for all conditions. Although the indication might depend on the institutions’ interest and no consensus or guideline currently exist, I inserted a paragraph for indications based on the context of this review.

[new, inserted at line 289]

PSG may be indicated for anatomically challenging sites [11,35, 57], the requirement of precise osteotomy in combination with PSI or MBA [6, 8, 35, 58-59], and the preservation of joint or bone stock with precise or complex osteotomy [21,22].

Reviewer 2 Report

The manuscript deals with a topic of great relevance with possible future applications in orthopedic surgery which can make oncologic major surgery easier and safer. The article is overall well written and remarkably innovative.

-In the introduction, the authors briefly and concisely described the fields of application of the 3D printing technology and the objectives of the study;

- In the next two sections, the authors thoroughly described the technique and its fields of application. The literature review was carried out meticulously and the references included are adequate. Strengths and limitations of this new technology were also properly analyzed.

-Conclusions drawn by the authors are in line with the results shown in the available literature.

-Images included clearly show the concepts described in the text and tables are correctly structured.

Author Response

Thank you for the thoughtful comments on our article. Based on the suggestions, we attempted to improve the quality of this review. Additionally, we obtained proofreading service for the grammatical check.

Reviewer 3 Report

The given review article is focused on three dimensional printing technology in the area of orthopedic oncology. Authors have highlighted different techniques for patient specific cutting guides. The report is presented in an effective and systematic manner. 

Author Response

(The authors gave the same response as above.)

Reviewer 4 Report

This review summarizes bone tumor resection using patient-specific 3D printed cutting guidance. It is a comprehensive and informative summary of the latest findings in this area. It also describes in detail the problems with this technique, such as preparation time, cost, and learning curve. A validation study of the method is also presented, and the extent to which this method is reliable as a surgical aid is also discussed, which is worthy of evaluation. I believe that this paper is appropriate for publication in this journal.

Author Response

(The authors gave the same response as above.)

Reviewer 5 Report

Interesting article on a hitherto unknown topic. My comments: Abstract well written. Introduction: please add information that 3D printing is also used in planning surgical treatment of complex fractures and reconstruction of deformities. Methods well presented and readable. Results Well presented. Figures and tables are legible. Please do a broader review of the literature related to 3D printing. Good conclusions.

Author Response

“Interesting article on a hitherto unknown topic. My comments: Abstract well written. Introduction: please add information that 3D printing is also used in planning surgical treatment of complex fractures and reconstruction of deformities. Methods well presented and readable. Results Well presented. Figures and tables are legible. Please do a broader review of the literature related to 3D printing. Good conclusions.”

Thank you for reviewing. As you say, a broader topic on 3D-printing technology is important. However, we focused on the usage of 3D-printing technology for oncology treatment in accordance with the interests of readers of Current oncology and special issues. Anyway, I briefly mentioned the planning of surgical treatment of complex fractures and reconstruction of deformities in the introduction section.

[line 32-38] 

Three-dimensional (3D) printing technology has been introduced in the field of orthopedic surgery, including its application in total joint arthroplasty, the treatment of malunion, and the reconstruction of bone defects due to trauma or bone tumors [1-3]. Treating complex comminuted fractures requires extensive surgical experience and anatomical knowledge, and patients with malunion could require osteotomy to correct the deformity and relieve pain [4]. 3D printing technology could help precise and effective fracture fixation and reconstruction [4].

Meng M, Wang J, Sun T, Zhang W, Zhang J, Shu L, Li Z. Clinical applications and prospects of 3D printing guide templates in orthopaedics. J Orthop Translat. 2022 May 13;34:22-41. doi: 10.1016/j.jot.2022.03.001. PMID: 35615638; PMCID: PMC9117878.